# Microfabricated Nitinol Stent Retrievers with a Micro-Patterned Surface

**DOI:** 10.3390/mi15020213

**Published:** 2024-01-31

**Authors:** Shogo Kato, Yuzuki Ban, Takashi Ota, Norihisa Miki

**Affiliations:** Department of Mechanical Engineering, Keio University, 3-14-1 Hiyoshi, Kohoku-ku, Yokohama 223-8522, Japan; shogok110812@keio.jp (S.K.); yuzuki.ban@keio.jp (Y.B.); takashi94@z3.keio.jp (T.O.)

**Keywords:** stent retriever, nitinol, microfabrication, sputtering, thrombectomy, blood clot, in vitro, thrombus

## Abstract

Stent retrievers are medical devices that are designed to physically remove blood clots from within the blood vessels of the brain. This paper focuses on microfabricated nitinol (nickel–titanium alloy) stent retrievers, which feature micro-patterns on their surface to enhance the effectiveness of mechanical thrombectomy. A thick film of nitinol, which was 20 µm in thickness, was sputtered onto a substrate with a micro-patterned surface, using electroplated copper as the sacrificial layer. The nitinol film was released from the substrate and then thermally treated while folded into a cylindrical shape. In vitro experiments with pig blood clots demonstrated that the micro-patterns on the surface improved the efficacy of blood clot retrieval.

## 1. Introduction

Thrombus formation is the leading cause of acute cerebral infarction [1,2]. Early treatment is crucial, and rtPA, a medication that functions as a clot-busting drug, is administrated to stroke patients within 3 h of onset, and no contraindications [3]. Within the next 24 h, blood clots are physically removed using thrombectomy devices [4,5,6]. Stent retrievers are among the most effective devices due to the short procedure time and favorable clinical outcomes [7,8,9].

While other materials, such as stainless steel [10] and biodegradable polymers [11,12], are commonly used for indwelling stents that remain inside the vessels for the long term, nitinol is widely employed in the manufacturing of stent retrievers. Nitinol is an alloy of nickel and titanium and is well known for its superior properties, including its shape memory, super elasticity, and biocompatibility [13,14,15,16,17,18,19,20]. Conventionally, nitinol stent retrievers are manufactured using methods such as knitting nitinol wires [21,22,23], laser-cutting a tubular nitinol film [24,25,26,27,28,29,30], or laser-cutting a nitinol sheet and folding it into a stent retriever form [31].

In order to enhance the thrombectomy efficiency, it is reasonable to equip the stent retrievers with micro-patterned surfaces. However, the wires used for these stent retrievers have diameters within the range of several tens of micrometers, making it extremely challenging to add micro-patterns to them. Laser cutting, while effective, can produce burrs on the surface [32], necessitating subsequent electrochemical etching to remove them [27,28,29,30]. However, this electrochemical etching process also removes pre-patterned micro-features on the surface. Although holes have been patterned onto the surface after electrochemical etching [23,26], structures with convex surfaces, which are considered more effective than concave structures in thrombectomy, have not yet been reported. To the best of our knowledge, this is the first experimental investigation of thrombectomy while using a stent with microstructures.

The sputtering of thick nitinol films and their application to form cylindrical stent-like shapes was reported. Nitinol sputtering was applied to either a cylindrical substrate [33,34,35] or to a flat substrate, followed by thermal treatment, with the shape defined using a cylindrical mandrel [36,37]. The nitinol film needs to have a thickness of several tens of micrometers. This thickness is too large for other metals; huge residual stress would be induced, and the film would self-crack. However, nitinol, which is several tens of micrometers in thickness, can be formed by sputtering due to its unique properties [38,39,40]. Since this process does not require electrochemical etching, micro-patterned surfaces can be achieved either by micro-patterning the substrate surface in advance or by micro-patterning the sputtered film. However, the efficacy of thrombectomy with nitinol stent retrievers that feature micro-patterned surfaces has not been discussed.

In this paper, thrombectomy efficacy was investigated with respect to the micro-patterns of the surfaces. First, the fabrication process for the nitinol stent retrievers with micro-patterns was developed, and the stent retrievers were optically investigated. Second, the thrombectomy efficiency of the manufactured stent retrievers was experimentally assessed through in vitro experiments with pig blood clots. The impact of the surface micro-patterns is discussed later.

## 2. Materials and Methods

### 2.1. Fabrication Process of Nitinol Stent Retrievers with Surface Micro-Patterns

Figure 1a,b describe the specific dimensions of the stent that is expanded into 2D and folded to create 3D structures in Figure 1a,b, respectively. The size was determined with reference to commercially available stent retrievers. The stent is designed to have surface features to augment thrombectomy’s efficiency. Figure 2a,b detail the design of the surface features, including the striped and circular patterns.

Nitinol is the most frequently used material for stent retrievers due to its biocompatibility and super elasticity [21,22,23,24,25,26,27]. Particularly for medical use, the proportion of nickel in nitinol must fall within the range of 53.5% and 57.5%, according to the associated regulations [41]. In our study, we utilized a nitinol sputtering target with a nickel content of 55.7% and a titanium content of 44.3%, which was then applied in order to form the stent material.

The fabrication process of the nitinol stent is schematically described in Figure 3. The glass was chosen as the substrate material, taking into consideration the thermal expansion coefficient. The thermal expansion coefficients of glass, copper, and nitinol are 8–9, 16–16.7, and 6.6–11 [10^−6^/K], respectively [15,42]. Since the sputtering process involves heating, it is crucial for the thermal expansion coefficients of the materials to be similar. It is worth noting that when silicon substrates were used, cracks in the nitinol layer were frequently observed due to thermal stress.

Chromium (0.03 µm in thickness) and copper (0.1 µm in thickness) were vapor-deposited on a glass substrate (Micro slide glass S9112, Matsunami Glass, Osaka, Japan) as the contact metal and the seed layer for the subsequent electroplating, respectively (Figure 3a). Next, a positive photoresist (AZ4903, Merck Chemicals, Tokyo, Japan) with a thickness of 2.5 µm was patterned to form a mold for the subsequent electroplating process (Figure 3b). The conditions for electroplating are described in Table 1. A copper layer (1 µm in thickness) was formed via electroplating, and then the photoresist was removed (Figure 3c). Another positive photoresist layer (AZ4903), with a thickness of 25 µm, was patterned into the shape of the stent retriever (Figure 3d), with a width of 100 µm. Using this photoresist as the mold, copper was electroplated until it overflowed the mold (Figure 3e). When the photoresist was removed using acetone, the copper was overhanging the cavity (Figure 4), which is a crucial aspect of the final release of the nitinol layer (Figure 3f).

A nitinol layer (20 µm in thickness) was sputtered using a DC sputtering machine (BM-600, Shibaura Mechatronics Corporation, Kanagawa, Japan), as illustrated in Figure 3g, under the conditions described in Table 2. Since sputtering generates heat, in order to protect the sputtering machine, the process consists of 16 cycles of 5 min of sputtering with a 30 min interval. The sputtering rate was determined with reference to previous studies [34,36]. Additionally, it was reported that a higher sputtering power results in better step coverage, which is not preferable for the release process. Therefore, we set the sputtering power and the deposition rate to 2.5 kW and 4 nm/s, respectively, even though our equipment could handle up to 5 kW.

The sacrificial layer of copper was removed with an ammonium-based etchant (CB-801Y, Meck Bright), and the nitinol layer, which had the shape of the expanded stent retriever (see Figure 1a) and micro-patterns on its surface, was released (Figure 3h). The released stent structure was manually folded and then inserted into a 6 mm diameter glass tube. The rationale for choosing this diameter is discussed later. In the inert gas environment, specifically argon gas, the temperature was ramped up to 650 °C for 15 min and maintained for 20 min. Subsequently, the heating was halted, and the stent was cooled down to room temperature over a period of 5 h. Finally, the stent was securely attached to a nitinol rod, which was 0.5 mm in diameter and 60 mm in length, using epoxy resin.

In this study, we prepared 3 types of stent retrievers as follows: those with no surface patterns, those with stripe patterns, and those with circular patterns, as shown in Figure 2.

### 2.2. In Vitro Experiments with Pig Blood Clots

The diameters of the blood vessels in the brain to which thrombectomy is applied range from 1 mm to 10 mm [43,44,45]. As an initial attempt to manufacture a stent retriever with surface features, a stent diameter of 5 mm was targeted. In practical use, the expanded diameter of stent retrievers is generally larger than that of the blood vessel. In this study, a stent sheath was not used to introduce the manufactured stent into the glass tube, and since the glass tube was rigid, unlike a blood vessel, a 6 mm diameter glass tube was used. In future studies, the use of flexible tubes with variable diameters and shapes will be explored to more accurately simulate practical treatments.

Thrombus retrieval experiments followed the protocol described in a previous study [46]. The experiments were conducted at room temperature. Immediately after the blood was collected from healthy pigs, approximately 0.4 mL of blood was introduced into a glass tube with a 6 mm diameter, which rested at room temperature for 6 h to facilitate the formation of blood clots inside the tube. The mass of the formed clots was defined as the mass of the introduced blood. The stent was manually inserted into the tube. After visually confirming that the stent was adequately inserted into the clot, it was slowly retrieved. The difference in the mass of the glass tube before and after its retrieval was used to estimate the mass of the retrieved blood clot. Thrombectomy efficiency was defined as the ratio of the mass of the retrieved blood clot to that of the originally formed clots inside the glass tube.

All of the experimental procedures were conducted by a single researcher. For each stent retriever, the experiments were conducted 8 times. After each experiment, the entangled thrombi were carefully removed manually. Subsequently, the stent retriever was rinsed in distilled water, and after confirming the physical removal of the thrombi, the stent was cleaned with ethanol and dried. Any evidence suggesting that the experiments affected the stent or its efficiency was not observed.

## 3. Results and Discussion

### 3.1. Fabrication of Nitinol Stent Retrievers with Surface Micro-Patterns

Figure 3 shows microscopic images and SEM images of the fabricated stent retrievers. As shown in Figure 5a, cylindrical stent retrievers were successfully formed, with a diameter of 5 mm, and were slightly smaller than the inner diameter of the mandrel. Figure 5b,c provide close-up views of the stents, revealing smooth edges that suggest electrochemical etching may not be necessary, which is crucial for maintaining surface micro-patterns. The width of the stent patterns was measured at 50 µm; this is smaller than the width that was determined by the electroplated copper (100 µm). The difference was attributed to the overhanging electroplated copper and the poor step coverage of the sputtering process.

Despite the poor step coverage and the overhanging structures, it was observed that nitinol sputtered onto the sidewall of the electroplated copper. This may have led to the failure of the release process due to bridging the nitinol structures forming the stent and those that were to be removed (Figure 3h). While the sputtering of nitinol on the sidewall was not perfectly prevented, it was thin enough to be quickly removed during the releasing process. Figure 5d shows the SEM image after nitinol sputtering, showcasing the thin layer of nitinol on the sidewall of the copper.

Figure 5e,f illustrate the surface micro-patterns of circles and stripes, respectively. These features were successfully incorporated into the stent surface. However, during the release process with the etchant CB801Y, the surface features were affected; the height was reduced by 0.5 µm, and the edges were found to be round.

### 3.2. Assessment of the Thrombectomy Performance through In Vitro Experiments

The images of the in vitro experiments are presented in Figure 6. Blood clots were successfully formed inside the glass tubes, as shown in Figure 6a. Figure 6b displays the fabricated stent retriever that is attached to a nitinol rod with glue. It was then inserted into the glass tube, and when extracted (Figure 6c), the blood clots became entangled with the stent (Figure 6d–f). The mass of the glass tube before and after retrieval was measured to determine the mass of the retrieved thrombus and thrombectomy efficiency.

Figure 7 illustrates the relationship between the thrombus mass and the thrombectomy efficiency concerning the stent retriever with no patterns, striped patterns, and circular patterns. We repeated the retrieval experiments eight times for each stent. The thrombus mass was found to be 295 ± 65 mg. Although the thrombectomy efficiency of the stent with the patterns appeared to be slightly higher than that of the stent without patterns, no significance was found between each device in the Man–Whitney U test. It was reported that efficacy depends on the size of the thrombus in thrombus aspiration tests with stent retrievers [47,48]. The thrombectomy efficacy was calculated for cases where the thrombus mass was above and below average. As shown in Table 3, a significant difference was found for thrombi with a large mass. The difference in patterns did not show significance. The red thrombus is soft and prone to tearing into pieces, leading to restenosis in cerebral infraction [49,50]. Though the size of the thrombus in cerebral infractions varies, stent retrievers with surface micro-patterns show promise, with a comparable thrombectomy efficacy to stents without patterns for small thrombi and a higher efficacy than those without patterns for larger thrombi.

### 3.3. Discussion

This study used blood clots that were formed immediately after collection from a pig. Thrombi associated with strokes are typically categorized into two types. The first type is the red thrombus, characterized by a high concentration of red blood cells and fibrin. These thrombi are physically soft and easily fragment, often leading to their dislodgment from the stent retriever during thrombus retrieval therapy [51]. The second type, known as the white thrombus, primarily consists of platelets and fibrin. White thrombi have the physical characteristic of being hard and resistant to fragmentation, making it challenging for them to dislodge during retrieval. Studies have reported that approximately 80% of thrombi removed from cerebral infarction patients are of the red thrombus type [52].

A method for producing a thrombus model involves adding fibrin and thrombin to blood with anticoagulants, as reported in several studies [53,54,55,56]. Stored blood can be utilized, and various types of blood clots can be created through centrifugation. However, it is challenging to confirm whether the formed clots possess the same properties as those that are formed inside blood vessels. Another approach for creating a thrombus model is to store fresh blood without anticoagulants in a glass tube, allowing it to rest for several hours at room temperature to form blood clots. This process was employed in this study, as well as in a previous study [46,48].

There are several other methods for generating thrombus models. One such method is the gelatin model, which involves mixing gelatin with milk, blood, or other substances. However, instead of mimicking the mechanical properties, this model is designed to replicate the visual appearance of a clot and is commonly used in ultrasound and photoacoustic imaging experiments [57]. Another approach is a blood clot model that uses skimmed milk and blood. While the mechanical properties of this model are closer to those of an actual thrombus when compared to the gelatin model, it remains challenging to accurately reproduce the deformation and viscosity of a thrombus [58].

Therefore, the blood clot used in this experiment is considered to be suitable for evaluating the stent retriever. Conversely, studies have reported that thrombi fragmentation during thrombectomy occurs more frequently at curved points in vessels than at linear points [59,60]. Additionally, as mentioned in Section 2.2, blood vessels are flexible, and the expanded stent retriever has a diameter that is slightly larger than that of the blood vessel during the retrieval process. Thrombectomy experiments using blood vessel models with different shapes and material properties should be conducted to bring new insights.

When dealing with large and heavy blood clots, a significant increase in thrombectomy efficiency due to the surface features was observed. While the stent retriever, featuring a mesh structure, captures thrombi by entangling them in vivo, thrombectomy experiments mimicking blood flow have reported a higher percentage of fragmentation with heavier initial thrombus weights [61,62].

The micro-patterns on the stent retriever, contributing to stronger entanglement and capturing forces through the increased surface area, appear to be particularly effective for larger and heavier thrombi. Further clarification of this mechanism needs be to pursued through experiments involving stent retrievers with a variety of surface features.

## 4. Conclusions

Nitinol stent retrievers with micro-patterns on the surface were proposed and successfully manufactured. Nitinol formed via sputtering does not produce any burrs and, thus, does not require electrochemical etching for smoothing. This allows the stent to have micro-patterns on its surface. Electroplated copper was used to determine the stent shape in the sputtering process. To minimize step coverage, the electroplated copper had overhanging structures. This was found to be critical in the following release process. The proposed manufacturing process for these stent retrievers with surface features offers flexibility in design modification through the adjustment of the mask design and film thickness during the copper electroplating process.

These experimental results demonstrate the effectiveness of stent retrievers with surface features in removing large and heavy blood clots, presenting a clinical advantage over conventional stent retrievers that lack such features. The proposed manufacturing process for these stent retrievers with surface features offers flexibility in design modification through the adjustment of the mask design and film thickness during the copper electroplating process. Conducting thrombectomy experiments using stent retrievers with a variety of surface features inside blood vessel models with different shapes and material properties could contribute to optimizing the stent retriever design.

## Figures and Tables

**Figure 1 micromachines-15-00213-f001:**
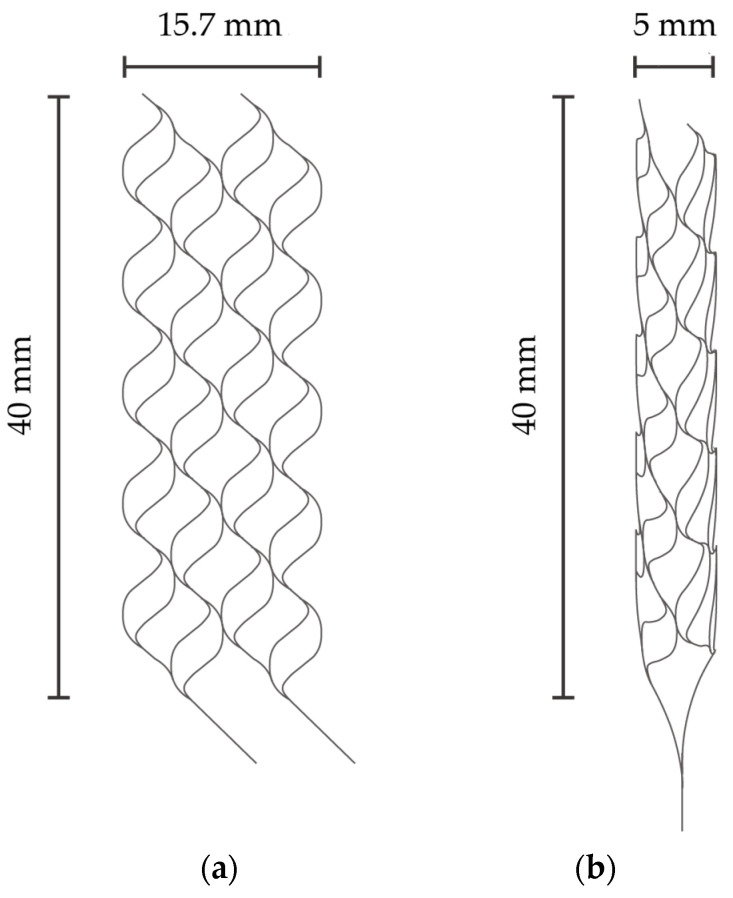
(**a**) Expanded stent retriever. (**b**) Three-dimensional stent retriever after thermal treatment.

**Figure 2 micromachines-15-00213-f002:**
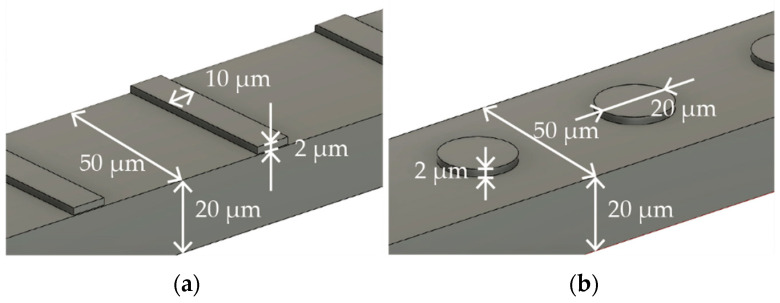
Surface features with (**a**) striped and (**b**) circular patterns.

**Figure 3 micromachines-15-00213-f003:**
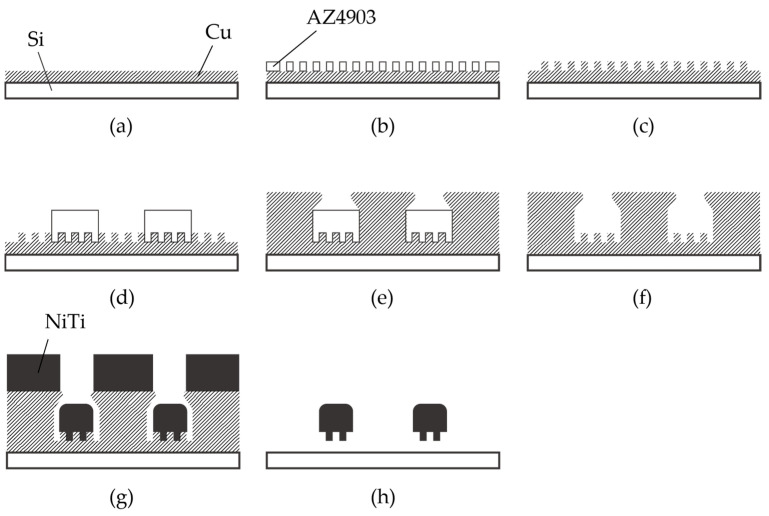
Fabrication process of nitinol stent retrievers using sputtering. (**a**–**c**) Formation of the mold used for surface micro-patterns. (**d**–**f**) Formation of the mold used for the stent structure. (**g**) Thick nitinol sputtering. (**h**) Released stent structure.

**Figure 4 micromachines-15-00213-f004:**
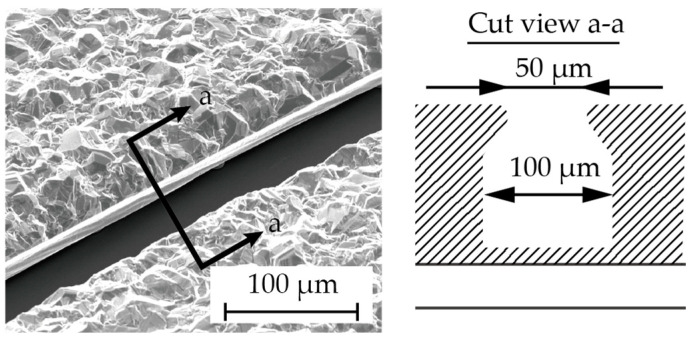
The cross section (a–a) showing the overhanging Cu structure in the fabrication step (Figure 3f).

**Figure 5 micromachines-15-00213-f005:**
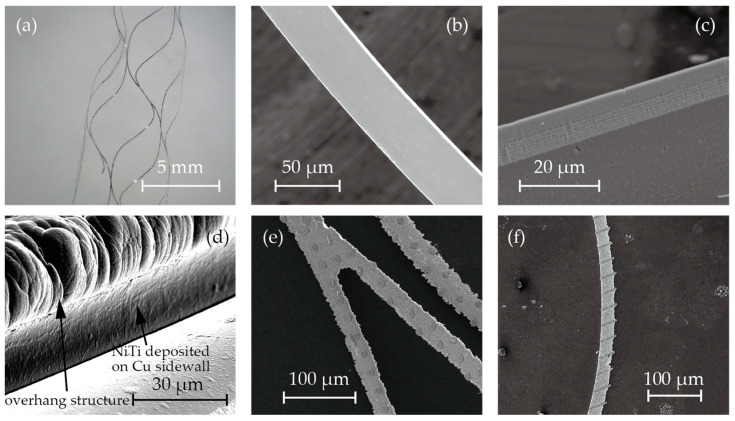
SEM photos of (**a**) cylindrical stent retrievers, (**b**,**c**) no-pattern stent retriever surface and edge, (**d**) nitinol deposited on overhanging copper sacrifice layer, and (**e**,**f**)circular and striped micro-patterns on the surface.

**Figure 6 micromachines-15-00213-f006:**
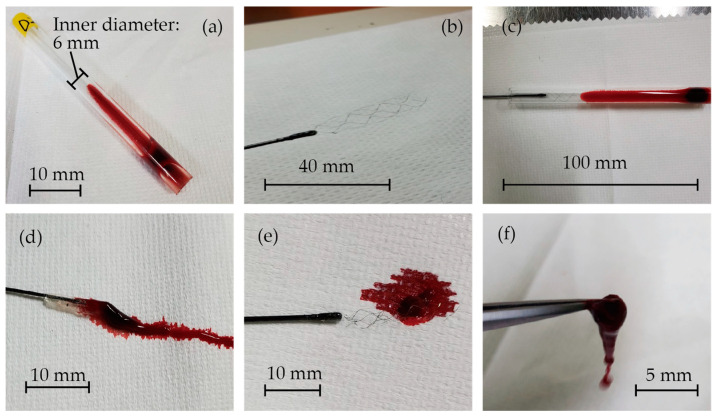
(**a**) Blood clots that formed inside the glass tubes, (**b**) the fabricated stent retriever attached to a nitinol rod with glue, (**c**) the stent retriever inserted into the glass tube, and (**d**–**f**) blood clots that were entangled with the stent.

**Figure 7 micromachines-15-00213-f007:**
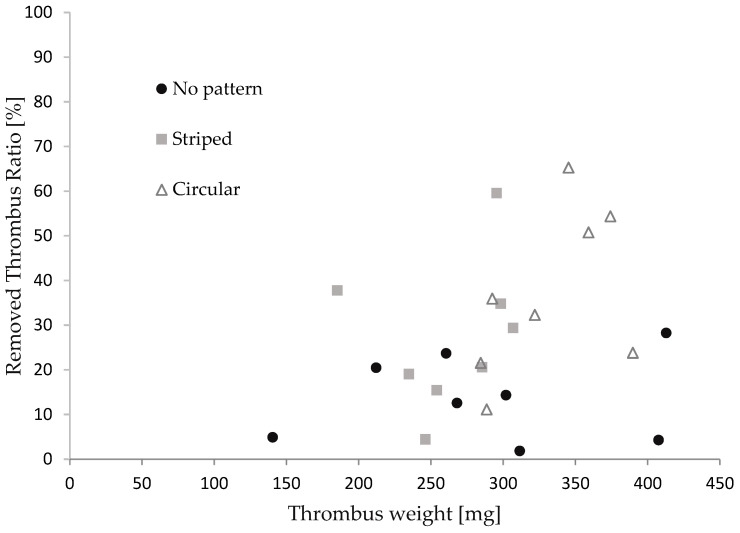
The relationship between the thrombus mass and thrombectomy efficiency.

**Table 1 micromachines-15-00213-t001:** Electroplating conditions.

Electroplating Conditions	
Plating bath	1 M, CuSO_4_
Current density	4.5 mA/cm^2^
Plating rate	70 nm/min

**Table 2 micromachines-15-00213-t002:** Sputtering conditions.

Sputtering Conditions	
Pressure	0.27 Pa
Power	2.5 kW
Ar flow rate	40 sccm
Deposition time	300 s
Interval time	30 min
Sputtering rate	4 nm/s

**Table 3 micromachines-15-00213-t003:** The results and significant difference.

Stent Retriever Type	No Pattern	Striped	Circular
Thrombus size	small	large	small	large	small	large
(n = 4)	(n = 4)	(n = 5)	(n = 3)	(n = 3)	(n = 5)
The average of removed thrombus mass [%] X ± SD	15.43 ± 8.43	12.19 ± 12.00	19.46 ± 12.02	41.26 ± 16.09	22.88 ± 12.46	45.32 ± 16.89
Significance (Mann-Whiteney)		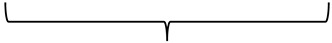 (*p* < 0.05)		
	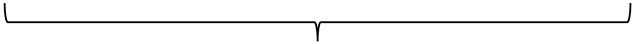 (*p* < 0.05)

## Data Availability

Data obtained during the in vitro experiments are found in the Appendix A.

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
