# Peer review of "Microfabricated Nitinol Stent Retrievers with a Micro-Patterned Surface"

_micromachines, 2024, doi:10.3390/mi15020213_

Round 1

Reviewer 1 Report

Comments and Suggestions for Authors

This paper developed a fabrication process for the nitinol stent retrievers with surface micropatterns. Laser cutting of nitinol films always followed by the electrochemical etching which does not allow the surface micro-features process. To manufacture the micropatterns on the surface of nitinol stent retrievers, the author proposed to form the nitinol stent by sputtering and to determine the stent shape by electroplated copper for minimizing step coverage. The proposed approach is quite interesting and impressive. However, there are some minor issues need to be considered.

1.     The manufacturing process in the "Material and Method" section needs more clarity. It's crucial to explain the rationale behind each step, material selection, and the control mechanisms used. Additionally, it is necessary to describe the specific dimensions of the stents manufactured in the article.

2.     For the description of experimental procedures in the “Materials and Method”, the decision to use a 6mm glass tube is unclear. The expanded diameter of the stent retrievers is generally larger than the diameter of the blood vessel. Is it reasonable for the author to choose a 6mm glass tube for the experiment? The ratio of the mass of the retrieved clots is better to descripted with a formula. A more detailed description of the experimental setup is necessary, such as environmental conditions, thrombus retrieval process, etc.

3.     The experiments were conducted for 8 times for each stent, however, is the stent used too many times? Will the stent affect the experimental results due to repeated use? This discrepancy raises concerns about the impact on results. Clarification or adjustment in methodology is recommended.

4.     The study presents only experimental results obtained under one single type of thrombus which might limit its applicability. Including 1-2 additional experiments with different thrombus compositions and consistencies could strengthen the findings and broaden the study's relevance.

5.     For the part of results, it seems that the analysis is vague and generic. The analysis is not in-depth enough, the author only described the phenomena observed without less discussion of how or why these phenomena happened.

6.     The conclusion succinctly summarizes the findings and their implications. Emphasizing the potential clinical impact of this research could strengthen the conclusion. Furthermore, it would be beneficial to provide a brief discussion on potential strategies for optimization and how it might influence thrombectomy efficacy.

Comments on the Quality of English Language

It is recommended that the author make a few grammatical corrections and standardize the language to improve the clarity of the article.

Author Response

We appreciate your review. All the revisions made are described in the attached file. Thank you for your time and effort for our paper.

Reviewer 2 Report

Comments and Suggestions for Authors

The authors have made a commendable effort in designing innovative stent retrievers utilizing the shape memory properties of Nitinol. To enhance the quality of the paper, I propose the following improvements:

1. Production Feasibility: On Line 35, it would be beneficial to providing references to support the challenges associated with microsurface printing would be valuable.

2. Literature on Alternative Stent Retrievers: If there is existing literature on other types of stent retrievers, it would be informative to include it. If such literature is not available, I recommend rephrasing the sentence for better clarity.

3. Sputtering Rate and Structural Fatigue: At Line 56, you mention a sputtering rate of 4 nm/s (I calculated from the deposition time). It would be insightful to include a discussion on the potential fatigue of the structure, given the high/low rate (compared with other references). Considerations of the impact of vacuum conditions during this process should also be mentioned.

4. Uniformity in Figure 3: Please ensure uniformity in terms of the scale bar in Figure 3, and remove any extraneous annotations from the optical microscope image.

5. Addition of FIB Image in Figure 4: Consider adding a Focused Ion Beam (FIB) image to Figure 4, as it would serve as a more robust external proof.

6. Clarity on Glass Tube Diameter: In Figure 5, explicitly mark the diameter of the glass tube, even though a scale bar is present.

7. As a suggestion: Curve Fitting in Figure 6: Explore the possibility of fitting the curve in Figure 6 with a suitable model. Also, investigate if there is a predictable correlation between the weight of the removed thrombosis and the thrombus for different designs. Some mathematical relations might give a better understanding of designs. 

Author Response

(The authors gave the same response as above.)
